# ENABLING SPARSE AUTOENCODERS FOR TOPIC ALIGNMENT IN LARGE LANGUAGE MODELS

## ABSTRACT

Recent work shows that Sparse Autoencoders (SAE) applied to large language model (LLM) layers have neurons corresponding to interpretable concepts. These SAE neurons can be modified to align generated outputs, but only towards **pre-identified** topics and with some parameter tuning. Our approach leverages the observational and modification properties of SAEs to enable alignment for **any** topic. This method 1) scores each SAE neuron by its semantic similarity to an alignment text and uses them to 2) modify SAE-layer-level outputs by emphasizing topic-aligned neurons. We assess the alignment capabilities of this approach on diverse public topic datasets including Amazon reviews, Medicine, and Sycophancy, across the currently available open-source LLMs and SAE pairs (GPT2 and Gemma) with multiple SAEs configurations. Experiments aligning to medical prompts reveal several benefits over fine-tuning, including increased average language acceptability ($0.25$ vs. $0.5$), reduced training time across multiple alignment topics ($333.6s$ vs. $62s$), and acceptable inference time for many applications ($+0.00092s/token$). Our anonymized open-source code is available in the attached zip file.

## 1 INTRODUCTION

A typical application of general-purpose LLMs is producing topic-specific generated text, also known as topic alignment. Existing approaches for topic alignment tend to use one of the following approaches: manipulating model input (e.g., few-shot learning, steering vectors, or prompt-tuning (Liu et al., 2023)), the model as a whole (e.g., fine-tuning, retraining (Bereska & Gavves, 2024)), or model output (e.g., output validation (Jarvis, 2023), regeneration, or filtering). As the costs (Weng, 2024; Kaplan et al., 2020) and interpretability challenges (Thirunavukarasu et al., 2023) associated with these existing approaches continue to scale (Villalobos et al.), they become impractical for applications that have multiple or changing alignment topics (Wu et al., 2024), that need some human control over the generation process, or that face precise, layer-level attacks (Mishra et al., 2024).

Recently, Sparse Autoencoders (SAEs) have been used as observational tools for LLM computations, and come from a set of tools in Mechanistic Interpretability (MI). When attached to an LLM layer, these SAEs decompose the layer output into SAE neurons corresponding to individual topics (e.g., the Golden Gate Bridge (Templeton, 2024)). Using SAEs as an MI approach for topic alignment can be promising because they provide:

**Computational Efficiency Once Trained** Precise alignment approaches make fewer changes to the model, so they are likely to be more efficient. As most LLM neurons encode multiple, unrelated concepts, known as *polysemanticity* (Bereska & Gavves, 2024), directly manipulating them could lead to vastly unexpected outputs, especially when multiple neurons are altered at once (Bills et al., 2023; Bereska & Gavves, 2024). Modifications at the next tier up in the *layer-level* outpu may be feasible and more efficient than the existing heavy-model editing alignment approaches.

**Increased Control** Recent research shows that SAE neurons can separate LLM layer outputs, which contain many concepts, into their corresponding human-interpretable concepts. Modifying these layer-level SAE neurons can align the eventual LLM outputs with more control than opaque approaches that may unexpectedly produce unaligned outputs.

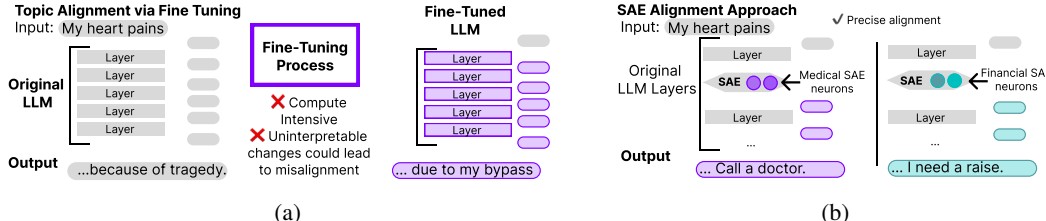

(a)                                                                     (b)

Figure 1: **(a)** shows how existing approaches make extensive changes to LLM weights during topic alignment, and **(b)** shows how our proposed approach aligns text to different topics by modifying aligned SAE neurons.

Although recent works have hypothesized that SAEs can be used for general topic alignment. (Gao et al., 2024; Templeton, 2024), and in doing so would be the first of the MI approaches used for downstream tasks, there are methodological gaps. First, to the authors' knowledge, there is no method to automatically identify alignment neurons relevant to a topic from the large potential set of SAE neurons **(M1)**. Second, no current methods manipulate SAE neurons for topic alignment in a context-sensitive way **(M2)**. Additionally, for SAE topic alignment to be a *viable* alternative to fine tuning, it must address current limitations. First, methods must be able to work with any SAE and LLM **(C1)** because SAEs have vastly different representational power (as we show in Fig. 3). Methods should also provide an uncertainty metric that quantifies the alignment modification across different tokens **(C2)**. Finally, methods should produce quality output without compute-intensive parameter tuning **(C3)**[1]. Accordingly, the main contributions of this paper enable SAE topic alignment by:

1. **Introducing the first methods** using SAEs for topic alignment by identifying & modifying SAE activations for **any** set of alignment topics without parameter tuning.

2. **Quantifying uncertainty** using a new metric that measures how aligned token output is.

3. **Performance evaluation** over multiple experiments across different SAEs configurations, three public topics datasets, including Amazon reviews, Medicine, and Sycophancy, and across the *only* open-source LLM-SAE pairs to the author's knowledge, GPT2 and Gemma. We observe promising results for topic alignment using correctness metrics, like increased language acceptability, and efficiency metrics, like reduced training time.

## 2 RELATED WORKS

While many related works surface alignment topics within generative models (including recent works like Chen et al.; Stoica et al. (2023)), few come from an interpretability subfield called mechanistic interpretability (MI). These MI methods focus on reasoning for neuron-level calculations and include:

**Logit-lenses and layer-level observational mechanisms:** Directly applying these observational MI approaches for topic alignment is difficult. Many samples could be needed before identifying possible alignment pathways, which would be impractical for general topic alignment tasks. SAEs have an advantage here because they uniquely act as both an observation and modification mechanism (Bereska & Gavves, 2024).

**Probing and modified vectors:** These layer-level approaches modify outputs towards a specific concept (e.g., 'weddings') (Zou et al., 2023; Han et al.; Turner et al., 2023). Approach limitations include the possible concepts represented, the uncertainty associated with the outputs, and the cost of training/verifying these probes across many concepts (Bereska & Gavves, 2024; Belinkov, 2021). Recently, Conmy & Nanda (2024) explored using SAEs to filter unaligned concepts from the steering vectors (by muting unrelated SAE neurons). This filtering approach could address some of the

---

[1]These methodological limitations and constraints are apparent while using the Gemma steering prototype from Neuronpedia (see Appendix: Fig. 7 ) (Neuronpedia, a).

trust limitations for steering vectors, but generalizability and uncertainty limitations remain. Nevertheless, Conmy & Nanda (2024) uncover a valuable insight – SAE modification using set values is very similar to steering vectors (Turner et al., 2023) because they both produce additive vectors for layer-level output.

SAEs are well-suited to address the limitations of other MI approaches and thus enable precise topic alignment. First, using SAEs with any model and any model layer is practical. There is already considerable investment in using SAEs across LLMs to observe the computational 'thought process' during token generation (Huang et al., 2024; Marks et al., 2024), and it would be efficient to reuse these SAEs for alignment as well. Second, SAEs learn different topics/concepts jointly (e.g., concrete nouns, syntax, more abstract concepts) instead of one at a time like modified vectors. Third, because the SAEs can be used at calculation within a layer, placing it at the multi-layer perceptron matches the intuition that alignment should occur (McDougall, 2023) where the model refines its output process for next token generation (Bloom, 2024). Fourth, the success of this method depends on the SAE representativeness, *not* the underlying LLM[2]. Given these benefits, this research aims to enable SAE for topic alignment by addressing the aforementioned methodological gaps and constraints.

## 3 METHODS

Our approach addresses methodological gaps in using SAEs (see Fig. 2a) for topic alignment by scoring how semantically similar SAE neurons are to an alignment topic and using those scores to select SAE neurons that contribute aligned output (see Fig. 2b). We also provide an uncertainty metric, *contamination*, quantifying the SAE modifications for output topic alignment.

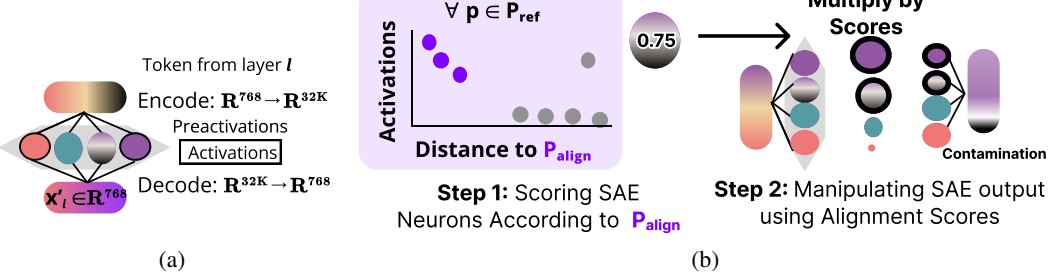

Figure 2: **(a)** SAEs take layer output, encode it into interpretable SAE neurons, and decode it back into a dense representation. **(b)** We calculate the similarity of each neuron in the hidden layer of the SAE to $\mathbb{P}_{align}$ and alter the token's activations proportionally.

**SAE Mechanics** As shown in Fig. 2a, SAEs process tokens. These tokens come from prompts $\boldsymbol{p} \in \mathbb{P}_{set}$ in a set of prompts. As tokens from a prompt, $\boldsymbol{p}_t$, pass through the model, they have a dense latent representation, $x(\boldsymbol{p}_t)$. At layer $l$, the SAE takes the dense representation input $x(\boldsymbol{p}_t)$ and encodes it into a higher dimension using encoding matrix $\boldsymbol{E}$ with dimensions $[d_{latent}, d_{hidden}]$. This encoding process disentangles the dense polysemantic representation with multiple concepts into one where each SAE neuron $i$ in the SAE's hidden layer, $\boldsymbol{h}_i$, should correspond to a single concept (Bereska & Gavves, 2024)[3] with a preactivation value: $\boldsymbol{\gamma}(\boldsymbol{p}_t)_i = (x(\boldsymbol{p}_t) \cdot \boldsymbol{E})_i$. Then, an activation function, $\sigma$, only selects some of these neurons to contribute to the final output: $\sigma(\boldsymbol{\gamma}(\boldsymbol{p}_t))$ (e.g. $\sigma$=Top-32k, where only the top 32 values from $\boldsymbol{\gamma}(\boldsymbol{p}_t)$ are nonzero). The non-zero post-activation neurons are decoded using $\boldsymbol{D}$ with dimensions $[d_{hidden}, d_{latent}]$ and the resulting dense token output, $x'(\boldsymbol{p}_t) = \sigma(\boldsymbol{\gamma}) \cdot \boldsymbol{D}$ is processed through the model.

---

[2]In theory, our proposed SAE approach works as long as the SAE with more hidden neurons than the dimension of the LLM embedding. In **all** cases today, as far as the authors know, the embedding is smaller than the smallest SAE we tested. To note, SAEs themselves have size limitations due to SAE training intensity.

[3]Multiple reports have demonstrated this disentanglement processes across different LLMs (Cunningham et al., 2023; Bricken et al.; Kissane et al., 2024).

Templeton (2024) use this SAE setup for steering. They clamp select SAE neurons' post-activation values high, and, because the contribution to the decoded dense representation is higher, that neuron's topic empirically has increased representation in the model output. However, using that approach for topic alignment more generally is complex because of the number of SAE neurons ($d_{hidden}$) and the semantic context that different neurons activate on. First, there could be multiple neurons that encode the same concepts with slight contextual nuances (e.g., Anthropic's neuron cluster for sycophancy (Sharma et al., 2023)) or neurons that need to co-activate for a desired output. Second, SAEs may have neurons that are still polysemantic and introduce some contamination into the output (some examples shown in Table 1.[4]). Notably, SAE neurons do not represent opposite concepts (e.g., truth and lie are separate neurons), so modifying SAE neurons can upregulate the presence of a topic, but clamping the neuron to 0 does not necessarily negate that topic. Thus, an automated method that scores SAE neurons by their semantic relevance could penalize both contextually irrelevant and polysemantic neurons without requiring manual SAE neuron identification for each alignment topic and token context. These scores can then be used to align tokens in a contextually-sensitive way so that alignment occurs on tokens that are easy to align vs. those that are not (e.g., syntax tokens).

| SAE Viewer | Examples of seemingly unrelated topics that activate same SAE neuron | |
|---|---|---|
| Gemma2b (Neuronpedia, b) | -based on **appellee** 's breach | -four **Gaelic** festivals |
| | -south side of the **chancel** are | -extensive **spectroscopic** coverage |
| GPT (OpenAI) | - fish-spawning **areas** | - transmission **lines** , pipes |
| | -expenses like **alimony** , payments | -alarm **clocks** or instant messengers |
| Claude (Anthropic) | - MeV) ¡" [* **Nu** cl. Phys.*] | -Spanish m **iqu** elitos |
| | - brett's Et **iqu** ette and | - joice in in **iqu** ity |

Table 1: Examples of polysemanticity, where a single neuron activates on unrelated topics, shown in **bold** (e.g., Gemma on tokens corresponding to astronomy, culture, and legal).

## 3.1 METHOD 1: SAE NEURONS SEMANTIC SIMILARITY TO ALIGNMENT TOPICS SCORES

With the premise that SAE neurons activate highly on only concepts related to $\mathbb{P}_{align}$ are alignment candidates. We could observe the SAE neurons that activate on tokens from $\mathbb{P}_{align}$ is a straightforward approach to identifying alignment neurons; we call this the *Strawman Approach*. However, given how few tokens $\mathbb{P}_{align}$ typically contains compared to the thousands of SAE neurons, there is no way to determine how polysemantic an activated SAE neuron is or identify similar neurons that would also be useful for alignment but did not activate on tokens.

Our proposed approach addresses these challenges by using a large reference set, $\mathbb{P}_{ref}$ with many concepts or topics. By independently processing prompts in $\mathbb{P}_{ref}$, each SAE neuron 1) likely has at least some tokens it activates, and prompts in $\mathbb{P}_{ref}$ that are similar to $\mathbb{P}_{align}$ can be used identify 2) other relevant alignment neurons. This baseline can then be used to identify and penalize SAE neurons activated with tokens as follows:

- **Aligned** neurons have high activations only when prompts are close to $\mathbb{P}_{align}$ and should be used for alignment.

- **Polysemantic** neurons have prompts close and far activate high.

- **Unaligned** neurons have prompts that are far and activate high and should be avoided.

All SAE neurons fall in one of these three categories, and polysemantic should be considered proportional where they fall between aligned and unaligned. Thus, for a score to quantify if an SAE neuron only activates highly on concepts related to $\mathbb{P}_{align}$, we want to penalize neurons that fire

---

[4]This limitation is explicitly expressed by Gao et al. (2024), "A large fraction of the random activations of [SAE neurons] we find, especially in GPT-4, are not yet adequately monosemantic."

highly on distant prompts. However, because SAE activations occur at the token level and distances are calculated at the prompt-level, we need both a prompt-level activation calculation and prompt-level distance from $\mathbb{P}_{align}$.

**Summarizing Prompt-Level Activations**  Tokens that pass through the SAE activate specific neurons, but prompts contain a variable number of tokens. Prompt-level activations should correspond to the *semantic* relevance of an SAE neuron to $\mathbb{P}_{align}$, so our approach normalizes the sum of neuron activations over all tokens and outputs a vector of prompt-level activations per SAE neuron:
$\textbf{summary}(\boldsymbol{p}) = \sum_t \sigma(\boldsymbol{\gamma}(\boldsymbol{p}_t))/g$ where $g = \sum_i^{d_{hidden}} \sum_t \sigma(\boldsymbol{\gamma}_i(\boldsymbol{p}_t))$

**Calculating Prompt-Distances**  Prompts of different lengths can be compared using sentence-level embeddings ($e$), to summarize prompts into a single vector, and a distance metric ($dist$), to calculate distances between vectors. The output prompt-level distance for a prompt in $\mathbb{P}_{ref}$ is the minimum $dist$ to any element in $\mathbb{P}_{align}$: $\min \{ dist(e(p), e(p')) \, \forall \, p \in \mathbb{P}_{align}, p' \in \mathbb{P}_{ref} \}$

To distinguish between aligned, polysemantic, and unaligned SAE neurons, we use the weighted variance equation (NIS, 1996), where $\mathbb{E}[dist] = 0$ for a perfect neuron.

$$g(h_i) = \frac{\sum (summary(p)_i * dist(p, p'))}{\sum summary(p)_i} \, \forall \, p' \in \mathbb{P}_{align}, \forall p \in \mathbb{P}_{h_i} \tag{1}$$

$$score(h_i) = \frac{(g(h_i) - \min(g(h_i)))}{(\max(g(h_i)) - \min(g(h_i)))} \tag{2}$$

Higher scores (between 0 and 1) for SAE neuron $h_i$ means increased relevancy to topics in $\mathbb{P}_{align}$[5].

### 3.2  METHOD 2: MODIFYING SAEs WITH ALIGNMENT SCORES

The alignment scores calculated in Sec.3.1 for each SAE neuron can now be used to modify incoming tokens to different alignment topics in a contextually sensitive way.

**Clamping Approach (Baseline)**  Based on the insights from Conmy & Nanda (2024), forcing a specific feature to be clamped high, as. Templeton (2024) have previously done, is akin to creating a steering vector. We use this idea as a baseline, where the 5 SAE neurons with the highest scores are clamped to 10x their value (inspired by Templeton (2024)).

However, determining this clamping value is nontrivial and if it is too high or low, it produces garbled output. Instead of parameter tuning per application, our approach modifies $\boldsymbol{\gamma}(p_t)$ so that the SAE neurons selected post-activation are more aligned and still match the token's context.

**Swap Approach (Proposed)**  As shown in Fig. 2b, the SAE decomposes the layer-level token output into preactivation values. Weighting these values by alignment scores can change the SAE neurons selected after the activation function. However, large modifications to preactivation values can lead to garbled output. Instead, our approach uses the indices of the modified SAE neurons post-activation with the original preactivation values, as shown below, before the decoding step.

$$\sigma'(\boldsymbol{\gamma}(p_t)) = \mathbb{I}[\sigma(\boldsymbol{\gamma}(p_t)_i) \neq 0] \cdot \boldsymbol{\gamma}(p_t) \tag{3}$$

In contrast to vector-based steering and clamping, the additive vector in this proposed approach changes based on the token context. We can quantify how related the generated output is to $\mathbb{P}_{align}$ by multiplying final activations by 'unalignment scores', (1-scores), which we call *contamination* $= \boldsymbol{\gamma}(p_t) * (1 - score(h_i))$. Through these methods, we address existing gaps in identifying SAE alignment neurons (**M1**) and using them to modify output (**M2**) while meeting the constraints because these methods are SAE agnostic (**C1**), lend naturally to the contamination uncertainty metric (**C2**), and do not rely on parameter tuning (**C3**).

---

[5]($g(h_i)$) can be used to compare SAE representation power across different alignment topics.

## 3.3 SAE Preliminaries and Implementation Details

As open-source SAEs (Gao et al., 2024; Lieberum et al., 2024) have only recently been released, there is little exploratory work on the representational power of different SAEs, a prerequisite for alignment.

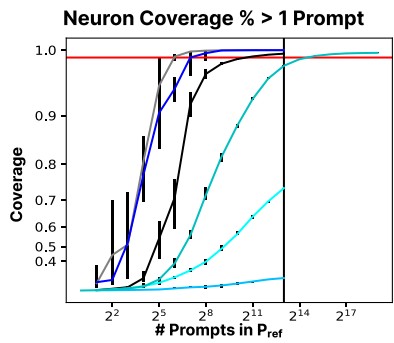
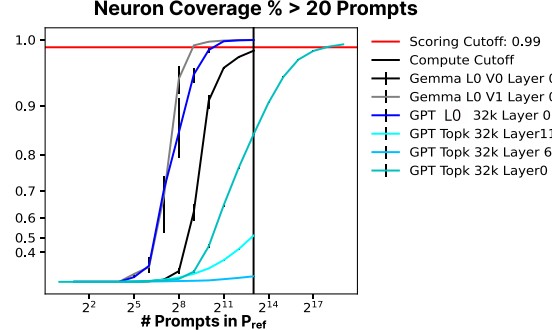

(a) % SAE neurons activated.

(b) SAE neurons % with $> 20$ prompts activated.

Figure 3: Comparing different SAEs and the coverage of different neurons across different sizes of $\mathbb{P}_{ref}$. Error bars represent the max and min values across 5 random samples of that size.

Our $\mathbb{P}_{ref}$ construction samples 1500 prompts across 660 tasks in HuggingFace's P3 dataset (Sanh et al., 2021) to form a pool of nearly 1 million prompts spanning different topics. For the prompt embedding $e$, we used sentence transformer (Reimers & Gurevych, 2019), and for $dist$, we used Euclidian distance.

To study SAE representation ability, we sample 5 $\mathbb{P}_{ref}$ sets with different sizes (with replacement) and exclude the <endoftext> token from each prompt so that they are not overrepresented in generated outputs. As seen in Fig. 3, the number of neurons activated varies widely by SAE Configurations, including the underlying model and layer, activation function ($\sigma$), and number of neurons in the hidden layer. Notably, most smaller layer 0 SAEs cross above 0.99 coverage around only $2^{14} \approx 8K$ prompts, whereas other SAEs from different layers have less coverage for the same number of prompts. These less representative SAEs likely have more neurons that only activate under rare circumstances or are dead (Gao et al., 2024; Templeton, 2024). If a neuron does not have enough prompts activated on it, we cannot adequately identify how polysemantic/unaligned it is, so we do not consider neurons where less than 20 prompts have been activated.

## 4 Experimental Results and Analysis

Our evaluation focuses on the 1) SAE neuron scores, 2) layer output, and 3) full model-generated output. We highlight results on the medical datasets because the presence of domain-specific terminology is an indicator of topic alignment. See Appendix for additional ablation studies over:

- **SAE Configurations** LLM model (GPT, Gemma), Layer of LLM, Parameters in SAE, SAE loss function/parameters, # of neurons in SAE.
- **Score Calculation**, specifically design choices for prompt-level summaries.
- **Alignment Texts** Topics - (P3: Amazon (Sanh et al., 2021), Medical (Gamino), Shoes ( Generated), Sycophancy (Rimsky et al., 2023)) Format - (6 prompts, 20 prompts).

## 4.1 SAE Neuron Alignment Scores

Our validation approach for scoring (**M1**) is inspired by the SAE neuron scoring *strawman approach* in Sec 3.1. Recall that $\mathbb{P}_{align}$ is never directly used during neuron scoring due to polysemantic and multiple similar neurons. By reversing this approach and calculating how many of a subset of the top$-k$ scoring SAE neurons also activate on $\mathbb{P}_{align}$ vs. an unaligned dataset (HuggingFace, b), we no longer face challenges due to polysemanticity and small token sample.

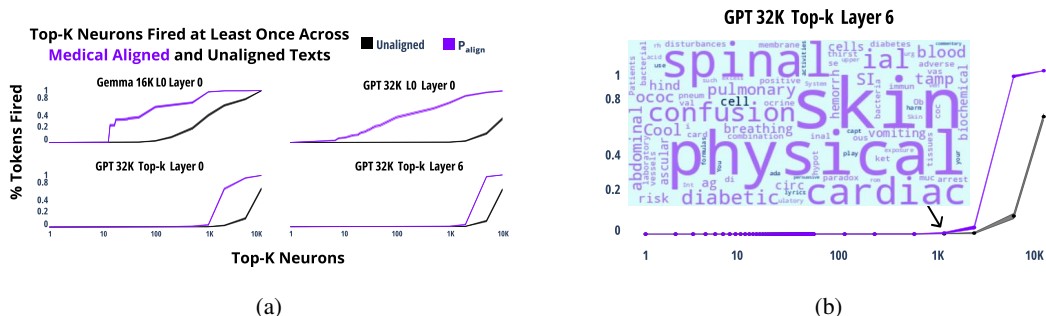

(a)                                                                          (b)

Figure 4: Top$-k$ evaluation for medical prompts. In **(a)**, we observe that aligned text (purple) generally activates higher than the unaligned text (black) on the top$-k$ neurons. **(b)** shows tokens that activate on the top$-k$ highest scoring neurons are generally words associated with the alignment topic.

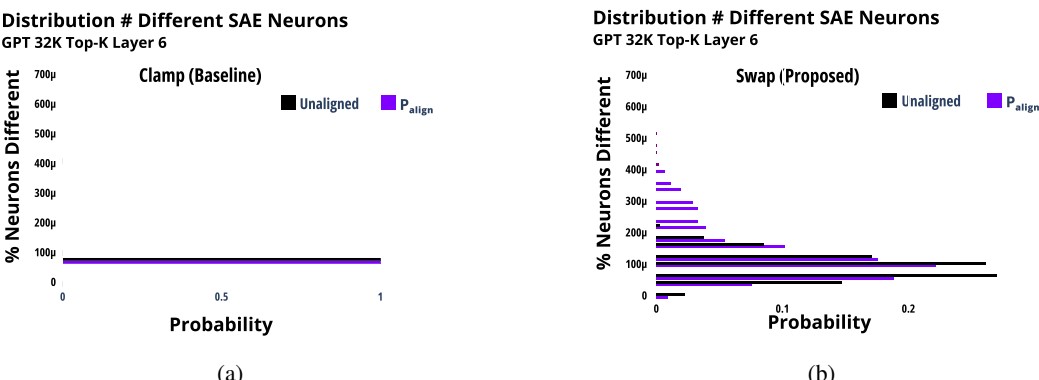

(a)                                                                          (b)

Figure 5: **(a)** shows that Clamp is not sensitive to incoming token context. While we can observe in **(b)** that Swap varies the number of neurons changed, which reflects that it accounts for token context and feasible alignment potential.

Results in Fig. 4 show that, as expected, a higher percentage of tokens from $\mathbb{P}_{align}$ (the medical dataset) activate on top$-k$ scoring neurons than the unaligned text. The value of $k$ at where the differences between the datasets become apparent (greater than 0) varies with the representative power of the SAE (Fig. 4a). Further, the tokens in either text, which are firing on the top$-k$ neurons, are still topically aligned (Fig. 4b), showing that the top-scoring neurons are similar to $\mathbb{P}_{align}$.

## 4.2    LAYER-LEVEL OUTPUT

Validating the immediate output of SAE modifications using alignment scores (**M2**) involves comparing Clamp and Swap methods with the unmodified SAE (Fig. 5). First, at a neuron level, Clamp has a static number of different neurons as we always force the changes for 5 neurons, while Swap has a distribution. Fig. 5 also shows that Swap makes more neuronal changes when the text is already aligned. This could be because having alignment scores between 0 and 1 yields changes in the post-activation SAE neuron set only when candidate neurons have high preactivations and high scores vs. unaligned tokens having high preactivations on SAE neurons with lower, non-zero scores, so the Swap multiplication does not change the postactivation SAE neurons selected.

Second, for layer-level modified outputs, we consider the following metrics, as shown in Fig. 6:

- **Difference in Reconstruction Error** (↓) Difference between the reconstruction error of the modification output (Modif vs. Orig) and the SAE output (SAE vs. Orig). Values less than 0 mean that the modification more closely matches the original token than the SAE.

- **Contamination** ($\downarrow$) As described in Sec. 3.2, it is a function of the post-activation neurons and measures modification misalignment.

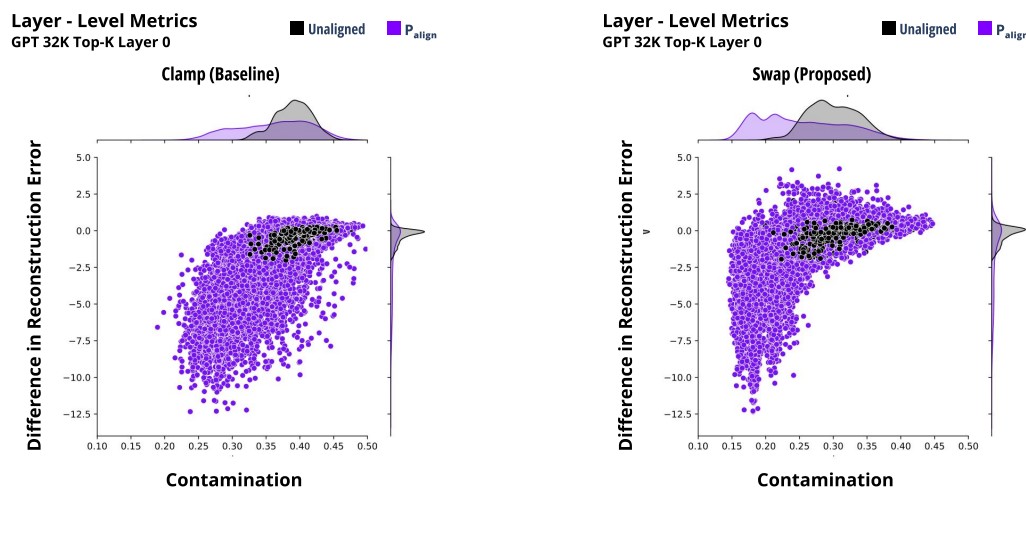

(a) Metrics for Clamp Approach.       (b) Metrics for Swap Approach.

Figure 6: Generally, the difference in reconstruction error for the modifications is less for aligned tokens than unaligned tokens. In Swap **(b)**, we see lower contamination as a metric of uncertainty than with Clamp **(a)**, which means that the neurons firing have higher alignment scores with Swap than Clamp. Special tokens not visualized.

In Fig. 6, the unaligned text has a higher difference in reconstruction error because the the more generic SAE trained to reduce reconstruction error should better represent the unaligned text while the SAE modifications should better represent the aligned text. We have excluded special tokens from our analysis, which had very high reconstruction error with our modifications. We hypothesize this is because we removed these special tokens during our scoring process, but they were included while training the underlying SAEs. This could be why the tokens with the lowest/highest differences in reconstruction error are the same between both mechanisms. Still, in the Swap approach, we tend to notice lower contamination (x-axis in Fig. 6b), which is in line with our expectations that Swap is better able to align layer-level output based on our alignment scores.

## 4.3 GENERATED OUTPUT

Lastly, we evaluate the full model outputs of our modification method (**M2**) to a fine-tuning approach using the following metrics:

- **Perplexity** ($\downarrow$) This is a standard metric of unexpectedness for next token generation, also used by Conmy & Nanda (2024); Turner et al. (2023).
- **Linguistic Acceptability** ($\uparrow$) A measure of how acceptable the generated text structure is using pretrained models trained for this task (Warstadt et al., 2019).
- **Distance from** $\mathbb{P}_{align}$ ($\downarrow$) Using prompt-distances described in Sec. 3.1 to determine min distance of the generated text from $\mathbb{P}_{align}$.

We compare our approach to an out-of-the-box fine-tuning method, adapted from HuggingFace (a), and generated 64 tokens per prompt with a top-k value of 5. To demonstrate alignment power, our input prompts were common medical metaphors (see Appendix for full prompts), as we expect that the general-purpose LLM would generate text related to the metaphor, while the alignment approaches should generate medical text. Our results in Table 2 show that approach performance depends on the underlying SAE's representational power:

| | **FT** | **Orig.** (Ground Truth) | **MI Approaches** | | | |
|---|---|---|---|---|---|---|
| | | | **Layer** | **SAE** | **Clamp** (MI Baseline) | **Swap** |
| Example Text: "My heart is broken, so I should" | laryngoscopy to visualize my heart. Iused cricoid balloon tubes to visualize the chest. I also used cricoid balloon tubes to visualize my stomach. | so thanking my friends. My heart breaks so my heart breaks [..] keep me safe and healthy in our | 0 [Good] | just tell my husband and wife to get out. [..] My kids have a daughter and she is a big sister and I am so jealous I will take a divorce. | be,.)- is, = ] is.and] is] on is ] =]is ] is a)] to get [ so. so [ is.]].] and was a lot] on the] in the in ons and]]'] | should go to sleep. It's not liek my brain is doing any work t oget me back up and working again. My head has stopped |
| | | | 6 [Poor] | I should be, I was-c-c ""R and then he was the ""C-I was-r, I was R R (I was)-P R (The ""R R R-T) The first is-C) "' | ?I I-3-in–, it? I... ""L?, ""L, ""......,, ""I?,, ""A I I I?..., I-A?, I I?, ""A?I-A......,I-I,,,? | not the ""c and I have it-f, the ""x is a very-al "" ""-and it-r-R R and a very) R-. (I was not the two) |
| Perplexity | 7.7e+36 ± 8.5e+36 | 4505 ± 1283 | 0 | 4767 ± 1440 | 136.2 ± 51.2 | 7416 ± 2635 |
| | | | 6 | 495.2 ± 322.3 | 126.7 ± 16.3 | 649.5 ± 98.0 |
| COLA | 0.25 ± 0.26 | 0.42 ± 0.29 | 0 | 0.75 ± 0.26 | 0.08 ± 0.16 | 0.5 ± 0.30 |
| | | | 6 | 0.00 ± 0.00 | 0.00 ± 0.00 | 0.00 ± 0.00 |
| Distance | 1.22 ± 0.05 | 1.300 ± 0.05 | 0 | 1.315 ± 0.042 | 1.353 ± 0.022 | 1.286 ± 0.060 |
| | | | 6 | 1.332 ± 0.015 | 1.352 ± 0.013 | 1.364 ± 0.018 |
| Contamination | N/A | N/A | 0 | - | $0.542 \pm 3e-5$ | $0.250 \pm 2e-5$ |
| | | | 6 | - | $0.427 \pm 5e-5$ | $0.307 \pm 2e-5$ |

Table 2: Standard metrics in SAE layers with different representativeness (0 is better than 6 see Fig. 3). Our results show that the standard perplexity metric is flawed (high values for FT in the medical domain), the MI baseline consistently returns outputs with low linguistic meaning, the swap approach is competitive on distance to alignment text (see Appendix for more details), and the contamination score determined by the topic alignment process aligns with the output quality on inspection.

Further, based on our generated text results in Table 2, we note that there are better metrics for alignment than perplexity, especially in domains with jargon. The fine-tuned approach has very high perplexity because the next predicted token in medical text is highly unexpected (and likely suffers from some numerical instability). Another reason this metric is limited is because it favors Clamp, which tends to only produce meaningless, repetitive text, as reflected by the Linguistic Acceptability (COLA) metric. Also, the distance from $\mathbb{P}_{align}$ is smallest using fine-tuning, but Swap, when the underlying SAE generates meaningful text (Layer 0), is second. Finally, the contamination metric provides some confidence that the Swap method relies more heavily on SAE neurons with higher alignment scores than the Clamp approach. These results demonstrate the potential of using SAEs, **given underlying SAEs that are highly representative.**

### 4.4 COMPUTATIONAL COSTS

Finally, the computational costs of this proposed approach support that it is worth investigating as a practical alternative to fine-tuning. Most of the training cost is a one-time set-up cost, as shown in

Table 3. Our implementation used $\mathbb{P}_{ref}$ with 8K tokens and $\mathbb{P}_{align}$ with 800 tokens that is broken down in Table 4. Further, while fine-tuning appears quite efficient at inference time (Table 3), there is an overhead because our implementation relies on packages like Transformer Lens instead of natively implementing PyTorch hooks. This is demonstrated by the gap between inference times for the SAE Original approach and the Fine Tuning approach. Thus, when comparing Swap and Original, there is a difference of 0.059s to generate 64 tokens. Still, while Clamp appears to take more time, we attribute this to implementation choices converting between types as an optimized version should take less time than Swap.

| Compute | Fine Tuning | SAE Approaches | | | |
|---|---|---|---|---|---|
| | | Original | SAE | Clamp | Swap |
| **Set-Up** | N/A | All are 12.4m | | | |
| **Per Task** | 333.6s | All are 62s | | | |
| **Prompt Inference** | $0.399 \pm .0009$ | $6.204 \pm 0.017$ | $6.235 \pm 0.019$ | $6.639 \pm 0.023$ | $6.263 \pm 0.014$ |

Table 3: Computational costs for training/inference across approaches.

| Task Breakdown | Type | s/Per Token | SAE Approaches |
|---|---|---|---|
| Ref Embeddings | **Set-Up**, Parallel | $0.07\pm0.001$ | 10m |
| Ref Latent Generation | **Set-Up**, Parallel | $0.02\pm0.001$ | 2.4m |
| Align Embeddings | **Per Task**, Parallel | $0.07\pm0.001$ | 56s |
| Distance Generation | **Per Task**, Sequential | $3e^{-4} \pm 1e^{-6}$ | 2.4s |
| Scoring | **Per Task**, Sequential | $4e^{-4}\pm3e^{-5}$ | 3.72s |

Table 4: Compute breakdown for SAE Set-up and Per Task in Table 3.

## 5 DISCUSSION AND CONCLUSION

This work enables topic alignment using SAEs by proposing new methods to address gaps. These methods involve calculating alignment scores for each SAE neuron and modifying the SAE outputs in a contextually-sensitive way with no parameters. With a competitive correctness performance and computationally efficient inference-time modification that takes less than $0.001s/token$ on average, the proposed approach is promising due to the interpretability properties of the SAE, quanifiable uncertainty, and lack of parameters. By unlocking topic alignment using SAEs, this work enables using SAE alignment as a tool to study other interpretability questions and use in applications where alignment topics change often. These results inspire research directions closely tied to exciting interpretability challenges like:

- **Designing** $\mathbb{P}_{ref}$: To reduce one-time compute costs, is there a better way to design $\mathbb{P}_{ref}$ using contrastive approaches? How can we design it so that potential $\mathbb{P}_{align}$ topics will be very close to some $\mathbb{P}_{ref}$ prompts and very far away from others?
- **Engineering SAE modification:** We show SAE representational power varies by configuration. Which layers should use SAE steering for alignment and what does that tell us about LLM self-repair (McGrath et al., 2023).
- **HAI Perspective:** This approach uses a large volume of data modifications, and users may benefit from an approach like Cho et al. (2024) to see how our methods change the outputs.

Further, research into improving the representational power of the underlying SAEs, testing on rare alignment topics, and studying patterns of coactivation can further the potential of using SAEs for topic alignment as an alternative to fine tuning. Finally, it is worth noting that this approach can also address other limitations with fine-tuning (e.g., where privacy is essential, as $\mathbb{P}_{align}$ tokens are never directly used for scoring, or when there are too few examples for successful fine tuning) and warrants exploration on these dimensions as well.

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

# A APPENDIX

**Reproductibility Statement:** Please find our open-source code in the attached zip file, implemented in PyTorch. Additional implementation details are provided in Sec 3.3. Experiments were conducted using NVIDIA V100 Tesla GPUs. For the synthetic "Shoes" dataset, we prompted an LLM with, "Generate 20 prompts related to shoes, separated by a comma", which we post-processed for syntax errors. All code and supplementary materials are released under the MIT License.

## A.1 ADDITIONAL JUSTIFICATION

As we focus on text generation, we do not compare with approaches related to other fine-tuning and alignment goals, such as high performance on a certain type of task (e.g. classification) or correcting how knowledgable, 'truthful', or factual models are, which have a rich body of supporting literature (Xu et al., 2023; Zhang et al., 2023; Hadi et al., 2023; Wang et al., 2023). Additionally, while there is continued exploration in identifying circuits of LLM neurons, that line of work is nascent and not ready for alignment applications without additional circuit discovery Tigges et al. (2024).

Specifically for alignment, we believe that existing approaches for alignment are limited, as shown in Fig. 7, and that the issues of polysemantic neurons and SAE representativeness are here to stay. First, there are far more concepts than the tens of thousands of hidden layer neurons in small SAEs, of which some 'dead neurons' do not activate on any large corpi of tokens Gao et al. (2024). Thus, many SAE neurons likely encode multiple concepts, especially in very rare cases. Increasing the width of SAEs is a popular research direction, but there are computational limits on training using current approaches. Additionally, if the SAE is too wide, it may learn concepts that are not present in the LLM, making resulting alignment mechanisms difficult.

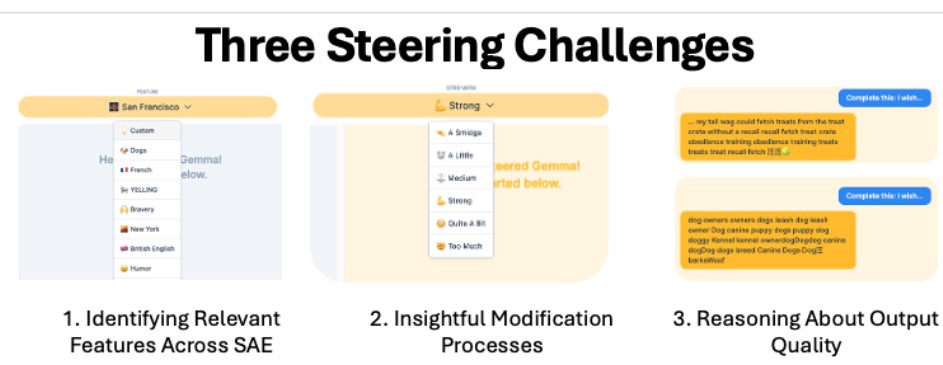

Figure 7: Recently released Gemma SAEs [6]Neuronpedia (a) can be used for steering at , but face three challenges emblematic of current research gaps our approach aims to address.

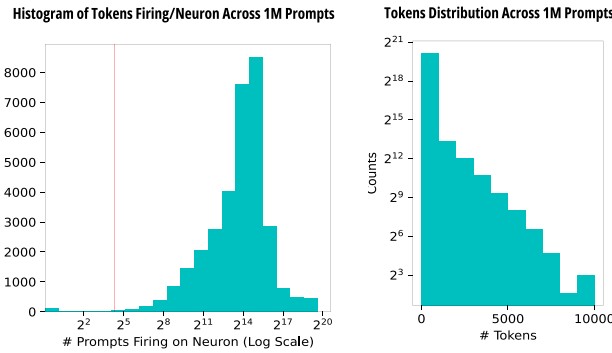

Figure 8: Most neurons activate on many tokens, but the number of tokens can vary widely across prompts.

As shown in Fig. 8, coverage is not uniform, with the vast majority of neurons activating on large numbers of tokens.

## A.2 DESIGN CHOICES FOR SCORING CONFIGURATIONS

For different combinations of SAE configuration and topic , we generated a score file, where each score summarizes prompt-level activations and distance graphs that look like the graphs in Fig. 9.

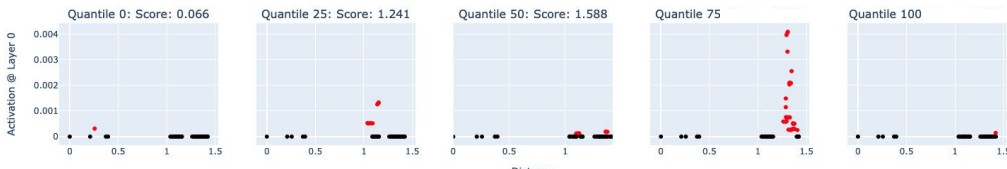

Figure 9: Examples of prompt-level activations across different SAE neurons in Top-k Layer 0, with a toy $\mathbb{P}_{ref}$ where n=2K (vs. 8K) for visualization purposes.

The respective violin plots of scores across the various configurations are shown in Fig. 10. Across models and topics, the score histogram's shape changes significantly due to SAE neuron coverage (see Fig. 3) and relevance. Across formats, we see that as there are more prompts, the scores shift up. This reflects that the more prompts there are, the smaller the min distance between prompts in $\mathbb{P}_{ref}$ and $\mathbb{P}_{align}$, which in turn increases the average score. This phenomenon reflects that there are more neurons that can be considered relevant to a larger set of $\mathbb{P}_{align}$. While the differences across prompt-level activation functions are difficult to visibly discern, we conducted a Kendall-Tau similarity test across combinations of the activations to discern how the relative ordering of neurons by similarity change with different distance functions, and across combinations this order is different (avg. Kendall Tau score 0.741).

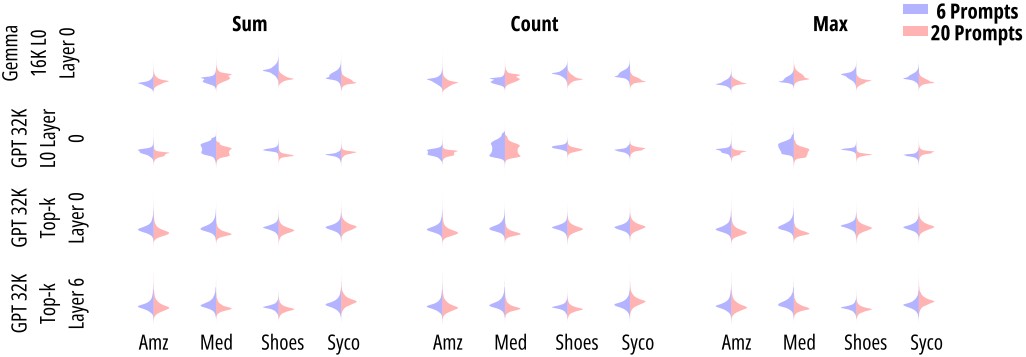

Figure 10: Scores across different listed configurations

**Choice of** $dist$: For four topics from the P3 set relatively individual tasks (Amazon, Yelp, IMDB, and Wikipedia), we sampled 20 prompts and calculated inter and intra-Euclidian distance metrics as shown in Table 5.

| Task | Text Excerpt | Compare | Inter-Topic Excerpt |
|---|---|---|---|
| Amazon | " I purchased these deck of cards[...]layout or spreads are too big, and it comes with a lil booketlet not really a booklet [...] do a reading for a man and women [...] Very disappointed" | Yelp | "I used to go here for tech books a lot. I went in and the section for tech books is half the size it used to be, completely disheveled, and contains no organization beyond the dummies books" |
| Amazon | " I guess there are times when the majority of the reviews do not live up to the movie. This movie was horrible to say the least. Horrible plot, acting, and props. Don't waste your time on this on, not even if you watch it for free." | IMDB | "[..]This is one of the most pointless films ever made. [...] Its a wonder that it was ever put on video.[..] Surely this film is a waste of the money used to create it, and a waste of anyone's time watching it." |
| Amazon | "" I guess there are times when the majority of the reviews do not live up to the movie. This movie was horrible to say the least. Horrible plot, acting, and props. Don't waste your time on this on, not even if you watch it for free." | IMDB | "Can only be described as awful. [...] impossible to believe that it can get worse - but fear not because it does.[...] script is worse than the acting or whether the directing is worse than both. [...] Lucky it only cost me $1 to hire. |
| Amazon | "This is a good read. A little slow at times. I recommend the entire Tucker Mills series. | Yelp | "ALWAYS very consistent. Food is ALWAYS good [...] So yummy" " |

Table 6: Distance Examples

| Task | Distances | |
|---|---|---|
|  | Intra-topic | Inter-topic |
| Yelp | 1.357 ±0.005 | 1.128 ±0.04 |
| Wiki | 1.412 ±0.004 | 1.144 ±0.041 |
| Amazon | 1.34 ±0.006 | 0.95 ±0.04 |
| IMDB | 1.34 ±0.005 | 1.00 ±0.04 |

Table 5: Distance Metric Evaluation

## A.3 SCORING OUTPUT

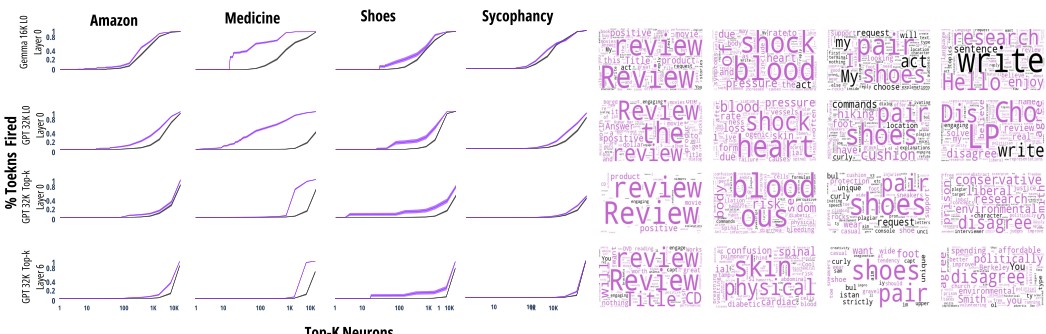

Figure 11: Additional Top-K Scoring Alignment Configurations (2 of 3)

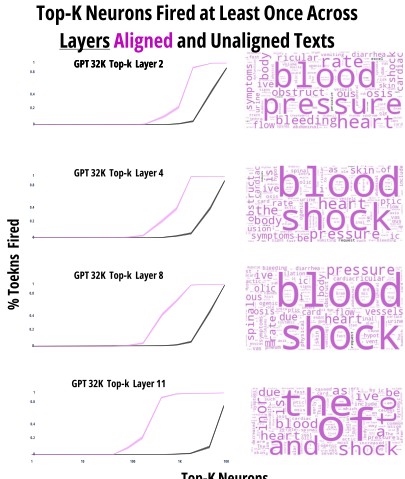

Figure 12: Additional Top-K Scoring Alignment Configurations (3 of 3)

We found the distance metric reasonable because the intra-topic distance was less than the inter-topic distance per topic. Still, some examples of prompts from different topics had a distance less than the maximum distance between intra-topic prompts. Upon inspection, these distances still made sense because even though they were from different tasks, they discussed similar topics, as shown in Table 6.

Generally, as shown in Fig. 11 and 12 $\mathbb{P}_{align}$ activated more on top scoring prompts than $\mathbb{P}_{unaligned}$, but this varied by the topic and model.

## A.4 LAYER-LEVEL OUTPUT

An ablation for the Swap method uses the SAE neuron alignment scores and multiplies them by the token activations without applying the masking step we use for Swap. However, this approach leads to high instability, as seen in the difference in reconstruction error in Fig. 13.

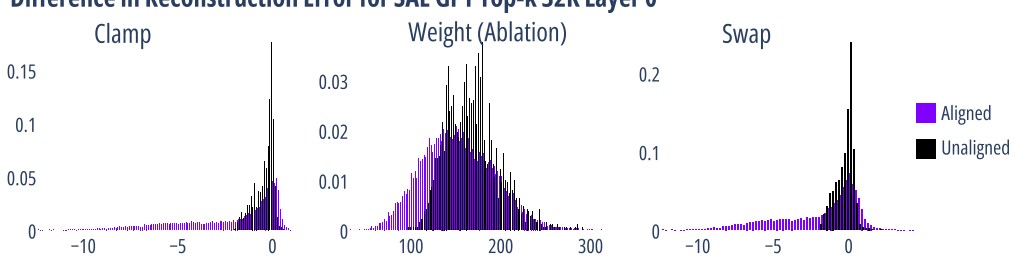

Figure 13: Results show an ablation of Swap without the mask and only using the new score-weighted SAE neurons (Weight). Unlike the other approaches, the range of difference in reconstruction error is far greater than 0, suggesting this is a poor modification mechanism.

## A.5 GENERATED OUTPUT

The prompts for the medical metaphor texts are:

1. What should I do about my heart of stone
2. What should I do about my heart pains
3. What should I do about tough skin

| Title | LORA + Prompt |
|---|---|
| Example | Input: "Please respond as a professional doctor: My heart is broken so I should not take this medicine." 
 Output: "I am not sure what to say or what to do next. Please don't ask. Please don't tell me I have been taken wrong. I have not. Please don't ask. Please don't tell me I am being taken." |
| Perplexity 
 COLA 
 Distance | $(11020 \pm 31)$ 
 $(0.5 \pm 0.4)$ 
 $(1.22 \pm 0.15)$ |
| Timing | 1. LORA Training Approach: $114\,\mathrm{s}$ 
 2. Inference 95% CI LORA: $(0.61\,\mathrm{s} \pm 0.22\,\mathrm{s})$ 
 3. Inference 95% CI LORA with Additional Prompt: $(0.90\,\mathrm{s} \pm 0.97\,\mathrm{s})$ |

Table 9: Metrics for the LORA + Prompt approach, including perplexity, COLA, distance, and timing as a supplement to Table 2.

4. How should I keep someone on their toes

5. Should I shoulder the blame for

6. Can I foot the bill

7. Do I have an eye for

8. My rule of thumb is

9. My heart is broken so I should

10. I have a cold so I should

11. I have a stiff upper lip so

12. I have butterflies in my stomach so I should

We provide additional correctness metrics for aligned prompts (from Gamino) and unaligned prompts (from HuggingFace (b)) in Tables 7 and 8.

| Aligned Text | | | | | | |
|---|---|---|---|---|---|---|
| **Metric** | **FT** | **Orig.** | **Layer** | **SAE** | **Clamp** | **Swap** |
| COLA | $0.25\pm0.19$ | $\mathbf{0.6}\pm0.22$ | 0 
 6 | $0.45\pm0.228$ 
 $0.0 \pm 0.0$ | $0.35\pm0.21$ 
 $0.0 \pm 0.0$ | $0.3\pm0.21$ 
 $0.0 \pm 0.0$ |
| Dist-ance | $1.13\pm0.05$ | $1.14\pm0.04$ | 0 
 6 | $1.11\pm0.03$ 
 $1.15\pm0.05$ | $1.15\pm0.04$ 
 $1.153\pm0.048$ | $\mathbf{1.11}\pm0.03$ 
 $1.15\pm0.04$ |

Table 7: Generated text using prompts from Gamino, which are aligned already.

| Unaligned Text | | | | | | |
|---|---|---|---|---|---|---|
| **Metric** | **FT** | **Orig.** | **Layer** | **SAE** | **Clamp** | **Swap** |
| COLA | $0.25\pm0.19$ | $\mathbf{0.8}\pm0.18$ | 0 
 6 | $0.45\pm0.2$ 
 $0.0\pm0.0$ | $0.1\pm0.1$ 
 $0.05\pm0.1$ | $0.35\pm0.2$ 
 $0.0\pm0.0$ |
| Dist-ance | $\mathbf{1.34}\pm0.03$ | $1.37\pm0.001$ | 0 
 6 | $1.36\pm0.01$ 
 $1.35\pm0.01$ | $1.35\pm0.009$ 
 $1.37\pm0.01$ | $1.37\pm0.01$ 
 $1.37\pm0.01$ |

Table 8: Results for different metrics across SAE layers

Finally, we show results for a LORA + Prompt tuning approach to supplement the standard fine-tuning approach (FT) in Tables 2-4:

