# OpenReview forum: "Enabling Sparse Autoencoders for Topic Alignment in Large Language Models"
_ICLR.cc/2025/Conference — Submitted to ICLR 2025_

### Official Review · Reviewer_Dvo9 · 2024-10-22

**Soundness:** 1
**Presentation:** 1
**Contribution:** 1
**Rating:** 1
**Confidence:** 5

**Summary:**

The paper's content is interesting, but unfortunately, the code contains the name of a person presumed to be the author. Therefore, I think this paper should be rejected.

**Strengths:**

The paper's content is interesting, but unfortunately, the code contains the name of a person presumed to be the author. Therefore, I think this paper should be rejected.

**Weaknesses:**

The paper's content is interesting, but unfortunately, the code contains the name of a person presumed to be the author. Therefore, I think this paper should be rejected.

**Questions:**

The paper's content is interesting, but unfortunately, the code contains the name of a person presumed to be the author. Therefore, I think this paper should be rejected.

---

> ### Author Response · Authors · 2024-11-13
>
> For anonymization, we used the standard anonymization platform (anonymous.4open.science) and included our names/affiliations in the regex to ensure it functioned correctly. Between submission and now, it seems the tool’s real-time performance may have changed unexpectedly. We've since removed our reliance on it.
>
> If possible, we'd greatly appreciate any feedback you might be able to offer, even from a general scientific perspective. We think it’s a promising idea and would value your insights.

---

> > ### Comment · Reviewer_Dvo9 · 2024-11-16
> >
> > The idea of this paper seems interesting. However, I am not sure why GPT-2, which is quite outdated, was chosen for the study. Would it be possible to demonstrate the results using more recent open-source LLMs?
> >
> > Additionally, the findings in Table 2 are not immediately clear. For example, it’s hard to discern whether the proposed method significantly outperforms other baselines, and I am uncertain about the specific point being made regarding contamination. It might be worth exploring better ways to present these results.

---

> ### Author Response · Authors · 2024-11-18
>
> Thank you for those comments! Hope this clarifies a bit.
>
> 1. **On GPT2**: To the best of the authors' knowledge, the only open-source SAEs released have been for GPT2 and Gemma so we’re limited by what is publicly available, but luckily the SAEs that have been released open source across a number of configurations include both ones that have high and low representation power (see Fig 3) which is the dimension most relevant to methods that rest on top of SAEs (that practitioners don't have control over).
>
> 2. To recap, our methods are the first to enable SAEs for fine-tuning. These SAEs are from a class of Mechanistic Interpretability methods, of which the natural baseline is a steering vector (Clamp). These mechanistic interpretability approaches modify output at the layer level of a model and can be quite precise so we can use their weights to quantify some form of uncertainty. Our proposed approach for this is called contamination and is not available in approaches that aren't from this class of mechanistic interpretability techniques.
>
> In light of this, our **results** aim to highlight the following:
> A. The success of our open-source methods fine-tuning depends on the underlying representation power of the SAE (which varies and usually depends on the company that trained it). In our results we gave an example of 2 SAEs at different layers, one with good representativeness and one with poor representativeness (output of 0 vs. 6). In this plot, contamination is the uncertainty represented by the **generation processes** under the hood (which we can only do because this approach is a mechanistic interpretability one).
>
> B. Some traditional metrics used in the literature for alignment like perplexity and distance may not be as informative in practice like we tried to demonstrate here. Perplexity is very low for the Clamp approach which produces nonsense but astronomical for Fine-Tuning, and distance in this medical setting allowed for nonsensical medical terms to be strung together from the fine-tuned text (see FT example), potentially exposing private information in medical settings vs. limited to the set of features represented in SAEs (usually common language words -- see Neuronpedia for more examples).
>
> Multiple reviewers have asked us to improve the presentation of Table 2: **Could we get your opinion on if presenting these takeaways directly on the table (e.g. in the descriptions on the left hand side or in the caption, or explicitly labeling 0 as poor representation SAE and 6 as good representation SAE)?**

---

> > ### Author Response · Authors · 2024-11-27
> >
> > Hello, I just wanted to send an update that it seems like most reviewers now agree on the content and we updated the manuscript with everyone's suggestions. In any case, please let us know if you have any additional questions or feedback as the review period comes to a close!

---

### Official Review · Reviewer_2Ym2 · 2024-10-26

**Soundness:** 2
**Presentation:** 3
**Contribution:** 2
**Rating:** 3
**Confidence:** 3

**Summary:**

This paper introduces a method for topic alignment in LLMs using Sparse Autoencoders (SAEs) to emphasize sparse layer activation relevant to specific topics without fine-tuning the parameters. By scoring sparse token activation results, the method is able to perform topic alignment without fine-tuning the parameters, reducing training time and enhancing interpretability. Experiments across diverse datasets demonstrate the method’s adaptability.

**Strengths:**

1. Novel approach for using sparse token encoding to do the topic alignment.
2. This work is highly relevant for applications requiring frequent topic shifts in generated text, offering a scalable and interpretable alternative to fine-tuning.

**Weaknesses:**

1. Poor performance. The method's performance remains suboptimal. While the authors present a novel approach to topic-alignment fine-tuning, the results showcased in Table 2 still lack clarity, with the distance score favoring traditional fine-tuning as the more effective option. In addition, providing a ground-truth case will make it easier to know what kind of output is desired. From the current cases, I believe all responses are not helpful for the users. Meanwhile, could the authors clarify what specific benefits their method might offer in comparison, such as interpretability or flexibility, that justify its use over standard fine-tuning except for the efficiency? Efficiency considerations are addressed further in Weak Point 4.
2. Lack of proper comparison. The paper lacks a comprehensive comparison with other common methods for topic-based fine-tuning, such as prompt hints, LoRA, and token-based fine-tuning. Given that a simple prompt hint typically provides a strong baseline, it is unusual that this comparison is missing. It would be helpful if the authors could add this baseline, alongside relevant metrics like computational efficiency, interpretability, and overall performance, to provide a fuller assessment of their method’s value relative to established alternatives.
3. Lack of proper target models and experiment details. The experimental setup lacks sufficient detail regarding the target models used. Although GPT-2 and Gemma are mentioned briefly, there is no information on model size, version, or specific configurations. To support a more robust conclusion, I recommend that the authors apply their method across a variety of large language models (LLMs) and provide clear details on each model's configurations including framework, version, and size. This would mitigate any potential bias introduced by the training data of a single LLM and enhance the generalizability of the findings
4. The cost analysis in the paper is incomplete, particularly regarding the training time associated with adding a sparse encoding layer. Since a sparse encoding layer is not a standard module for most LLMs, the additional time and computational resources required for training should be included in the analysis. A more detailed breakdown of these costs would give readers a more accurate understanding of the method’s overall efficiency and practical feasibility

**Questions:**

1. What LLM did you use for the experiment? In detail, what is the version and model size?
2. Have you tried using simple prompt hints? For example, please respond as a professional doctor or just use GPT-4 to rephrase the prompt to the desired domain.

---

> ### Author Response · Authors · 2024-11-16
>
> Thank you for taking the time to review this paper! We believe that we address many of these concerns within the paper.
>
> 1/2. On Performance and Evaluation
>
> In **Table 2**, we can observe that the distance metric here reflects the underlying representation ability of the SAE. While the fine-tuning approach could spit out many 6+ word medical terms (some of them not real), the **underlying SAE** did not have many of these  medical features. This is a benefit of our model's approach (and has strong implications for privacy) as we discuss -- see Table 7 for the *opposite* result on the distance metric on different prompts.
>
> The ground truth, in this case, is the “original” column - just the generated text without any modifications.
>
> We wanted to showcase that not all responses are beneficial. Our original premise was to enable sparse autoencoders for this task. SAEs have a lot more work to be done – and this is being done in many AI groups. The **outputs rely on the representation ability of the SAEs** and we tried to highlight this in our results by **showing our method across good and bad SAE layers** so that this highlights the benefits of our methods even in cases where the underlying SAE is poor via a high contamination score (which is important to researchers as the SAE is something that they may not have control over -- more in your 4th point).
>
> As far as other benefits, my background is in the healthcare domain so I’ll primarily talk from that perspective. In order to use LLMs in many spaces practically, there are challenges with liability. What inspired this work was that because SAEs have the unique property of both observation and modification, it has strong implications for liability/security if it can be used for fine-tuning. Here we show how it can be (with more details at the end of our discussion section). I would love to share this knowledge with the greater healthcare community and if you think adding more specific examples to that domain would help, that would be great to know.
>
> 3. The **only** two open source SAEs that have been released are for GPT-2 and Gemma as far as the authors know. While we put the configuration details in the figure texts (e.g., Fig 3), we can make this more explicit in a table. Our evaluation was structured to provide the greatest spread of results by design (e.g. one good SAE one bad), but we’re happy to add more experiments or move them up from the appendix across any dimensions you think we might have not been able to highlight particularly well. For more details, please see the response for Reviewer 2 for a worked out example on why **LLM size is currently not the limiting factor - it is the SAE size**. We will clarify this in the manuscript as was raised by several reviewers.
>
> 4. This paper is about **enabling already trained SAEs** (which the MI research community collectively has spent a lot of compute on) for **additional tasks in fine-tuning**. This paper is the **first** as far as the authors know to  show that we can develop methods so that Mechanistic Interpretability methods can be used for common ML tasks to *bridge important community gaps* detailed in [1], and [2], which were released after this paper was submitted. We can change the framing of our sentence on this to either bold it or add more details:  *“There is already considerable investment in using SAEs across LLMs to observe the computational ‘thought process’ during token generation (Huang et al., 2024; Marks et al., 2024), and it would be efficient to reuse these SAEs for alignment as well.”*
>
> Please let us know what you prefer!
>
> 1. For these experiments, we used GPT2 and Gemma 2 from their respective HuggingFace Implementations as they were the only reliable options for this type of evaluation as far as the authors know.
>
> 2. Although prompt hints are not a mechanistic interpretability approach and thus are not from the same class of methods as SAEs, adding them is quite straightforward and can add additional information to a reader deciding between the tradeoffs of using this type of mechanistic interpretability approach or a more standard one. We’ll update the tables in the next week or add them to the appendix accordingly.
>
> [1] “Is mechanistic interpretability about to be practically useful” S. Casper, Less Wrong
>
> [2] “Evaluating feature steering: A case study in mitigating social biases” Anthropic

---

> ### Comment · Reviewer_2Ym2 · 2024-11-16
>
> "This paper is about enabling already trained SAEs (which the MI research community collectively has spent a lot of compute on) for additional tasks in fine-tuning. This paper is the first as far as the authors know to show that we can develop methods so that Mechanistic Interpretability methods"
>
> If I understand correctly, the focus of this work is not on a novel application but rather on a Mechanistic Interpretability method. That said, I still struggle to see how this approach directly enhances the potential applications of LLMs. While it might make the models more interpretable and less of a black box, can you guarantee that the activation results in the sparse layer will always be explainable?
>
> Additionally, I’m curious about the differences between this approach and the one proposed by OpenAI [1]. The key point I want to emphasize is that many studies in this domain seem to be putting the cart before the horse—offering approximations without a mathematically solid foundation to ensure consistent, interpretable results. Without this, I find it hard to identify a truly groundbreaking distinction. Whether using sparse layer activations, standard attention maps, or even another LLM to observe patterns, all of these seem like potential approximations rather than definitive explanations.
>
> Ref:
> 1. Bills S, Cammarata N, Mossing D, et al. *Language models can explain neurons in language models* [J]. URL: https://openaipublic.blob.core.windows.net/neuron-explainer/paper/index.html. (Date accessed: 14.05.2023), 2023, 2.

---

> ### Author Response · Authors · 2024-11-19
>
> 1. I’ll start with a simpler question first –  “can we guarantee neurons will always be explainable?”
>
> **No** and automated LLM-based methods (and I, a human) fail to make sense of the examples in Table 1. In poor SAEs neurons can be polysemantic (encoding multiple concepts at once). The polysemanticity of neurons is important for our method. Otherwise the baseline Mechanistic Interpretability approach in Gemma steer (see Fig 7) where you take a handful of neurons and Clamp them high would be enough for fine-tuning (sans scalability limitations using Baseline Clamp).
>
> If it’s *interesting*, I’ll also answer a variant of your question as your question shows great intuition towards *Discussion Point 1*. In comparison to the black box, you’re right that our approach provides more information and while *guarantees* on generative models are certainly contentious, I want to share how we mitigate risks. Our scoring mechanism is based on a uniform sample of P3 prompts  and gives us a set of neurons which fire on over a cutoff number prompts each (e.g. ours in 20 see Fig 2b, so 0.2% of our P3 sample). As Appendix Fig 8 and the Fig 2 comparison shows, most of these neurons activate on a large number of tokens.
>
> While method inputs of P_ref and distance metric *depend on the practitioner*, given a representative random sample as input (P3 in our case) and a reasonable cutoff, the law of large numbers across these neurons supports the histogram profiles (distributions) that we use for scores in our evaluation (in line with work in 'normal profiles' for anomaly detection) -- which is why our experiments in the Appendix also consider in-sample and out of sample topics.
>
> 2) Thoughts on Bills
>
> This is great intuition and it is brought up in the Gao Sparse Autoencoder paper from OpenAI that came **after** the Bills paper (this is the one we use the GPT2 Autoencoders from), which I’ll copy below. From your comments, I can guess you might *not find the Bills **or** the Gao* approach convincing, so I’ll share my view. From everything I've read, it is clear that SAEs, *like most generative methods,  cannot have precise guarantees sans strict constraints and conditioning on specific domains* (e.g. theoretical followup work from discussion Point 1).
>
> Yet, applications do not necessarily need this type of guarantee. From a practical perspective, our application barrier entry was to show some more control than the black box approach (and circuit based approaches, if they do work, don't have the generality across topics as we bring up in our appendix). As discussed above and in Figs 2 and 8, the data-driven profiles across most neurons come from many prompts while we had a cutoff for neurons that rarely fire that we couldn’t profile well (ghost neurons). This type of empirical data profiling draws from a long history in the data mining community (e.g. anomaly detection and the conclusion over time that what constitutes a `normal' profile depends on the practitioner) that has been used in practice.
>
> So, when it comes to novelty, we still see this as an important contribution to two communities. The first is the the Mechanistic Interpretability community [1 from above] which is lacking methods like this and the second is to folks who still want to use the SAE approach in practice because it meets their needs.
>
> Very curious to hear your thoughts !
>
> *Excerpt from GAO about BILLS*: “Anecdotally, our autoencoders find many features that have quickly recognizable patterns that suggest explanations when viewing random activations (Section E.1). However, this can create an “illusion” of interpretability [Bolukbasi et al., 2021], where explanations are overly broad, and thus have good recall but poor precision. For example, Bills et al. [2023] propose an automated interpretability score which disproportionately depends on recall. They find a feature activating at the end of the phrase “don’t stop” or “can’t stop”, but an explanation activating on all instances of “stop” achieves a high interpretability score. As we scale autoencoders and the features get sparser and more specific, this kind of failure becomes more severe.
>
> Unfortunately, precision is extremely expensive to evaluate when the simulations are using GPT-4 as in Bills et al. [2023]. As an initial exploration, we focus on an improved version of Neuron to Graph (N2G) [Foote et al., 2023], a substantially less expressive but much cheaper method that outputs explanations in the form of collections of n-grams with wildcards. [... omitted experimental design due to length restrictions]  In general, autoencoders with more total latents and fewer active latents are easiest to model with N2G.”
>
> Gao, L., et al. "Scaling and Evaluating Sparse Autoencoders." arXiv, 2024, https://arxiv.org/abs/2406.04093.

---

> > ### Author Response · Authors · 2024-11-19
> >
> > To prevent additional reviewing load, if you are still curious about LORA + Prompt results, we've added them to Reviewer kE2w's thread.
> >
> > ---
> >
> > **Questions to the reviewer**
> > 1. Do you have any more questions/concerns about the *content* of the paper? We'd be happy to share additional information or experiments as needed. Please let us know so we have enough time to run them if needed.
> > 2. Else, we would like to make the *presentation* changes to Table 2 that other reviewers suggested would make our takeaways more clear. Please let us know if you have any additional clarifications you'd like us to make in the text!

---

> > ### Comment · Reviewer_2Ym2 · 2024-11-20
> >
> > Here are my thoughts:
> > Since the paper focuses on finding a better method for interpretability, I suggest including an experiment to quantitatively evaluate how your approach compares to other methods, such as the mentioned LLM-based activation method. In the original paper, the evaluation seems to emphasize application-oriented metrics, for example, comparisons with fine-tuning. While this highlights the utility, it falls short of establishing a truly novel contribution to interpretability.
> >
> > If you aim to claim a contribution to interpretability, it would be important to demonstrate that your method provides **better** interpretability rather than simply achieving some degree of interpretability. A well-designed experiment that quantifies and compares interpretability across methods would significantly strengthen the paper's argument.
> >
> > To summary, if you can provide proper experiment, I will raise my score.

---

> ### Author Response · Authors · 2024-11-20
>
> Thank you for being willing to raise your score!
>
> To clarify:  I would still keep the scope from the title, which is enabling SAEs (which historically have only been used for observational tasks via data profiling) for fine-tuning (topic alignment).
>
> So a critical difference between this and the **autointerp** work here is that auto-interp is **a way to explain the generation process [observational interpretability]**
>
> But **our approach** is **quantifying** the process by which we fine tune using the **manipulations that we make [an active process]** at the layer-level using SAEs.
>
> If we’re aligned on that there are some concerns with using the Bills approach, namely that this can't yet be used for steering in our scope. They’ve listed their monosemantic neurons here and they’re quite limited (100): https://docs.google.com/spreadsheets/d/1TqKFcz-84jyIHLU7VRoTc8BoFBMpbgac-iNBnxVurQ8/edit?gid=0#gid=0).
>
> *If you’ve seen an autointerp methods being used before for fine-tuning across a large number of topics (that are not our baseline of steering vectors in Clamp), please let me know because they haven’t come up in anything I've seen.*
> Let me know if I'm missing something...

---

> > ### Author Response · Authors · 2024-11-27
> >
> > Hello! As the review period is coming to a close we just wanted to ask if there are any other clarifications we can make. In light of our discussion, I’ve pulled up the description of the limitations of autointerpretability for LLM neurons or circuit-based observational techniques for fine-tuning from the appendix to the intro.
> >
> > Hopefully this change in the manuscript ordering clarifies our contribution of fine-tuning methods for SAEs as an application of an interpretability technique enabled by the **properties of SAEs to observe and modify layer-level output through the same mechanism**  *vs.* the class of observational methods like autointerpretabilty or more recently, circuits -- which to the best of my knowledge there are no general topic-alignment approaches for to compare like I did with the additional LORA + prompt experiment.
> >
> > Please let me know if you have any other suggestions or concerns!

---

### Official Review · Reviewer_UkkW · 2024-10-30

**Soundness:** 3
**Presentation:** 4
**Contribution:** 3
**Rating:** 8
**Confidence:** 3

**Summary:**

This paper presents a novel approach for using Sparse Autoencoders (SAEs) to enable topic alignment in Large Language Models (LLMs) without requiring computationally intensive fine-tuning. The authors propose two main methodological contributions: (1) a scoring mechanism to identify SAE neurons relevant to alignment topics, and (2) a modification approach that uses these scores to alter SAE layer outputs in a context-sensitive way. The paper demonstrates the effectiveness of their approach across multiple datasets (Amazon reviews, Medicine, Sycophancy) and models (GPT2, Gemma), showing improvements in language acceptability and training efficiency compared to fine-tuning approaches.

**Strengths:**

1. **Novel Approach**: The paper presents an innovative method for topic alignment using SAEs, addressing a significant gap in the literature. The approach is well-motivated by the limitations of existing methods like fine-tuning.

2. **Technical Depth**: The methodology is thoroughly developed with clear mathematical formulations for both the scoring mechanism and modification approach. The authors provide detailed justification for their design choices.

3. **Comprehensive Evaluation**: The experimental evaluation is extensive, covering:
   - Multiple datasets and topics
   - Different model architectures
   - Various SAE configurations
   - Both quantitative and qualitative metrics

4. **Practical Utility**: The approach shows promising results with:
   - Reduced training time (333.6s vs 62s)
   - Acceptable inference time overhead (+0.00092s/token)
   - Improved language acceptability in some configurations

**Weaknesses:**

1. **Limited Scale**: While the approach is tested on GPT2 and Gemma, there's no evaluation on larger, more current models. This raises questions about scalability.

2. **Parameter Sensitivity**: Though the authors claim their approach doesn't require parameter tuning, the results show significant variation across different SAE configurations and layers. More analysis of these dependencies would be valuable.

3. **Baseline Comparisons**: While fine-tuning is used as a baseline, comparison with other lightweight adaptation methods (like prompt tuning or LoRA) would strengthen the evaluation.

4. **Theoretical Foundation**: The paper could benefit from stronger theoretical justification for why the proposed scoring mechanism effectively identifies relevant neurons.

**Questions:**

1. How does the approach scale to larger models? Have you tested or analyzed computational requirements for models like LLaMA or GPT-3?

2. The results show significant variation in performance across different layers. Could you provide more insight into how to select the optimal layer for applying the SAE modifications?

3. How robust is the approach to different types of alignment tasks? While medical domain alignment shows promising results, are there certain types of alignment that are particularly challenging?

---

> ### Author Response · Authors · 2024-11-16
>
> Thank you for your review!
>
> 1. How does the approach scale to larger models? Have you tested or analyzed computational requirements for models like LLaMA or GPT-3?
>
> This concern came across multiple reviewers, so we will make sure to update the manuscript to clarify this point. To the best of our knowledge, the *only* open-source SAEs released so far have been only for  GPT2 and Gemma. However, even as the parameters per layer scale, the real limitation on this method the **size of the SAE not the LLM** (even today's LLMs with the highest dimensions 12k for GPT2 only needs an SAE with a dimension of 12k+1 in theory which is far less than the 16k neurons in our smallest SAE tested). (See more details to Reviewer 2 for more details if relevant!)
>
> 2) The results show significant variation in performance across different layers. Could you provide more insight into how to select the optimal layer for applying the SAE modifications?
>
> This is a great question and it relies on a particular assumption we want to clarify. What we show in this paper is that we can steer text, but that is **only as good as the underlying SAE**. So while Fig 3 shows that the released open source SAEs vary significantly by different layers, that could change depending on how one trains their SAEs (also an active area of research).
>
> What this means is that right now, because there’s still a lot of work to be done on the SAE side, we can't yet answer these questions. However, once SAEs do catch up and we find the right way to train them, it would go a long way not only in helping us ‘break the black box’ around LLMs – it is a future I hope this research area gets to explore!
>
> 3) How robust is the approach to different types of alignment tasks? While medical domain alignment shows promising results, are there certain types of alignment that are particularly challenging?
>
> **In our appendix you can find the results of a few other tasks!** The challenge once again lies with the SAEs that underlie this method. What we noticed across a few different topics(see Appendix Fig 11) is decreased performance on sycophancy vs. shoes, the medical domain, and Amazon reviews, but this reflects the SAE neurons present (e.g., if you have an entirely new concept that is not in the SAE, we can’t steer and luckily the uncertainty metric will inform that no steering occurred).
>
> To note: in accordance we are adding more baselines in our results from outside of Mechanistic Interpretability. While the clamp approach is the baseline for mechanistic interpretability v.s. other fine-tuning approaches like LORA which are outside of mechanistic interpretability, our main constraint was the size of the table and we'll try to work around that/rearrange some of the material to add those comparisons.
> For the theoretical foundations, we refer to the benefit this approach has over [Templeton 2024] in building up the field based on the polysemantic behavior we observe and the equations we base on it as a result.
>
> Please let us know if we can provide any more information!

---

> > ### Author Response · Authors · 2024-11-27
> >
> > Hello, I just wanted to send an update that it seems like most reviewers now agree on the content and we updated the manuscript with everyone's suggestions. In any case, please let us know if you have any additional questions or feedback!

---

### Official Review · Reviewer_ebxQ · 2024-11-02

**Soundness:** 2
**Presentation:** 2
**Contribution:** 3
**Rating:** 6
**Confidence:** 2

**Summary:**

This paper proposes using Sparse Autoencoders (SAEs) for topic alignment in LLMs to achieve efficient and interpretable topic alignment by scoring SAE neurons based on their semantic similarity to specific alignment topics. Authors demonstrate the effectiveness of the proposed method across datasets (e.g., medical and Amazon reviews) and models like GPT-2 and Gemma to show the advantage of SAEs in reducing training times and improving interpretability. The authors also introduce contamination metrics to quantify topic alignment uncertainty.

**Strengths:**

1. The paper introduces an interesting approach by applying SAEs to align LLMs with specific topics, which is relatively unexplored. Compared to FT, this approach is more interpretable and efficient.

2. This paper evaluates the proposed method across several datasets and models, showing the effectiveness and generalizability of the results. Further analysis, including the alignment score, SAE neurons distribution, and model generation output, covers various aspects of alignment performance.

3. The contamination metric is simple to evaluate the uncertainty in alignment.

**Weaknesses:**

1. The experiments are conducted on GPT2 and Gemma. It is unclear if the method is easy to use on other model families or larger models e.g. LLAMA 405B. It seems that the configuration would be hard to obtain on larger models.
2.  It seems that the configuration of SAE is varied on different topics.  Further investigating the impact of different configurations may be valuable.

**Questions:**

1. Does the proposed score/method evaluate the impact on the polysemantic neurons?
2. Is this method easy to scale with much larger LLMs with billions of parameters?
3. The contamination metric is used to quantify alignment uncertainty. Is there any human evaluation other judgment to show the reliably of this metric?

---

> ### Author Response · Authors · 2024-11-16
>
> Thank you for your feedback
>
> **1) Regarding the LLMs used**
> To the best of the authors' knowledge, the *only* open-source SAEs released have been for GPT2 and Gemma so we’re limited by what is publicly available.  However, one important thing to note is that even as the parameters per layer scale, the real limitation is the size of the SAE *not the LLM*.
>
> Example:  Right now, GPT4 has the largest dimension embedding that the authors know with a size of 12k. In theory, the SAE approach works as long as you have and SAE with more hidden neurons than the dimension embedding (12K+1). This is far less than the smallest SAE we tested (16K neurons)! And SAEs themselves are limited in size because they’re so difficult to train -- beyond a certain size it would take more training time than atoms than are in the universe by Anthopic’s estimate.
>
> **2) On the configurations of SAEs.**
> For this work, we use already off-the-shelf trained SAEs given that they were given to the research community to use  as a standard (e.g. with clear descriptions on how they were put together and on what datasets).  The provided code repository allows to run experiments over different dimensions of SAEs. Given the appx. 500 SAEs+, we structured our analysis across the most salient dimensions (e.g., version and layer) so that it can be contained in one paper, and the methods we build on top of them can showcase the potential of the proposed approach (v.s. the underlying SAEs).
>
> **Questions:**
> 1. Does the proposed score/method evaluate the impact on the polysemantic neurons?
>
> **Yes**, that is correct, it does. We take polysemanticity into account in our scoring through the trick of only evaluating how relevant a particular neuron is to a given topic (which is why this method is SAE configuration agnostic because differently trained SAEs are more susceptible to this phenomenon). You can see an example in the Appendix Fig 9 where the neurons at quantile 50+ are polysemantic, given that they activate (red) on different topics and are scored by their proportional relevance to the topic.
>
>
> 2.  Is this method easy to scale with much larger LLMs with billions of parameters?
>
> **Yes** Hopefully, the above example helps with distinguishing the scale between LLM size and the respective SAE used.
>
> 3. The contamination metric is used to quantify alignment uncertainty. Is there any human evaluation other judgment to show the reliably of this metric?
>
> We’re lucky that in mechanistic interpretability, the main theme is that the changes we make are the impact we have by design. While we didn’t do an IRB study/human evaluation, what’s exciting about this work is that this is the first time we’ve seen SAEs, which have been touted for their precise properties, be used in practice. For context, right now the evaluation state of the art is having LLMs evaluate LLMs (Templeton), which has major flaws over which this type of systematic evaluation is an improvement consistent with other alignment evaluations (Liu 2023).
>
> If you think we should change the framing for folks from other subdisciplines, we welcome any suggestions!

---

### Official Review · Reviewer_kE2w · 2024-11-05

**Soundness:** 3
**Presentation:** 2
**Contribution:** 2
**Rating:** 6
**Confidence:** 3

**Summary:**

This paper introduces an approach for achieving topic alignment in LLMs using Sparse Autoencoders (SAEs), offering an alternative to computationally intensive fine-tuning methods. The key contribution is a technique that scores SAE neurons based on their semantic similarity to desired alignment topics and then modifies layer outputs by emphasizing high-scoring neurons.

The methodology consists of two main components: 1) A scoring mechanism that evaluates SAE neurons' relevance to target topics using semantic similarity and a reference prompt set, and 2) A "swap" approach that modifies SAE outputs by emphasizing aligned neurons in a context-sensitive way. This enables topic alignment without extensive parameter tuning while maintaining interpretability through the SAE structure.

**Strengths:**

- The paper presents a new application of SAEs for topic alignment that addresses key limitations of existing approaches like fine-tuning in the topic alignment research area. The scoring and modification methods are generally well-motivated.
- The experimental methodology covers multiple LLMs, topics, and SAE configurations. The evaluation metrics cover both performance (perplexity, linguistic acceptability) and efficiency aspects.

**Weaknesses:**

My main concern is the lack of qualitative performance comparisons with conventional topic alignment methods, as indicated in the first paragraph of your introduction -- the proposed SAE-based method has advantages in computational efficiency, interpretability, etc. You should compare them. Additionally, the current organization of the paper is not self-contained; for instance, ablation studies should be rearranged to the main text. Regarding the motivation, justifications for evaluation metrics and methods in lines 200-206 (e.g., why these three aspects (aligned, polysemantic, unaligned) of neurons) should be clearly presented.

**Questions:**

See weaknesses. I suggest including more baseline results in the rebuttal phase.

---

> ### Author Response · Authors · 2024-11-16
>
> Thank you for your comments-- here are some clarifications from the paper to your comments.
> 1) On Evaluation
>
> Given that mechanistic interpretability is a newer subfield, the existing state of the art evaluation approaches are mainly having LLMs evaluate LLM output [Templeton and Gao] -- which is flawed. Our approach offers distinct benefits over the current state of the art and supports our initial claims across quantitative and qualitative metrics (Tables 2, 3 and 4).
>
> **Table 2** shows the output of the mechanistic interpretability approach of using SAEs is competitive with both existing fine-tuning approaches outside of mechanistic interpretability and a baseline mechanistic interpretability approach (Clamp), for distance and COLA (sentence meaningfulness) when the underlying SAE has *high representative power*. Our evaluation and presentation makes explicit the contributions of our method even when the underlying SAE has poor representation via a corresponding high contamination score.
>
> **Any other baseline would not come from Mechanistic Interpretability techniques (e.g. will not have a contamination score), but we are happy to add them to the appendix for context.**
>
> **Tables 3 and 4** clarify the applications for which this approach has promise in being computationally efficient. Additional correctness metrics for aligned prompts (from Gamino) and unaligned prompts in Appendix Tables 7 and 8.
>
> Specifically for interpretability, we carefully framed this paper as **enabling SAEs** (a tool for interpretability in LLMs that has generated significant interest across the community about their potential tasks) for topic alignment. This paper show how our methods bridge  increasingly important mechanistic interpretability gaps detailed in [1] and [2] – white papers that were released after we submitted this paper that show how important it this work is the the mechanistic interpretability subfield.
>
> [1] “Is mechanistic interpretability about to be practically useful” S. Casper, Less Wrong
>
> [2] “Evaluating feature steering: A case study in mitigating social biases” Anthropic
>
> 2) On structure
>
> To your point, the three types of neurons (aligned, polysemantic, unaligned) are important because all SAE neurons fall into one of these three categories. If they are aligned, they should be used directly and if they are unaligned, they shouldn’t be. What we show in **Table 1** is that, in practice, many SAEs have poor quality and may have polysemantic neurons that fall somewhere in the spectrum from aligned to unaligned. This practical insight inspires a scoring method v.s. the type of binary neuron selection we see in Gemma Steer (along with the other limitations in Fig 7 of that approach).
>
> *We greatly appreciate any help on the structure and organization of the paper to make these points more clear and that we find a consensus among the reviewers on the structure!*

---

> ### Author Response · Authors · 2024-11-19
> **Additional Results**
>
> ## Additional Experiments
>
> Here are additional results as requested. We emphasize that *these are **not** from the same class of approaches as our proposed* one using **SAEs** — only the **Clamp Baseline** is a congruent comparison in the context of this paper. Still, here are those results from using LORA to train the model and adding a prompt before. Given the results across the existing fine-tuning approach or original GPT2 are comparable to these results in Table 2 and still support our conclusions that Swap (Our Approach) is preferable to Clamp (Baseline) while still demonstrating why perplexity is a poor metric, we're curious where we should include them!
>
> ---
>
> *Example:*
> Input: “Please respond as a professional doctor: *My heart is broken so I should not take this medicine.*”
>
> Output: *I am not sure what to say or what to do next. Please don't ask. Please don't tell me I have been taken wrong. I have not. Please don't ask. Please don't tell me I am being taken.*
>
> *Perplexity*: \( 11020 \pm 31 \)
>
> *COLA*: \( 0.5 \pm 0.4 \)
>
> *Distance*: \( 1.22 \pm 0.15 \)
>
> *Timing*:
> 1. *LORA Training Approach*:  114s
> 2. *Inference 95% CI LORA*:  \( 0.61s \pm 0.22s \)
> 3. *Inference 95% CI LORA with Additional Prompt*:   \( 0.90s \pm 0.97s \)
> ---
>
> **Questions to the reviewers**
> 1. Do you have any more questions/concerns about the *content* of the paper? We'd be happy to share additional information or experiments as needed. Please let us know so we have enough time to run them if needed.
> 2. Else, we would like to make the *presentation* changes to Table 2 that other reviewers suggested would make our takeaways more clear. Please let us know if you have any additional clarifications you'd like us to make in the text and find some consensus. How can we help researchers scanning the paper find the information they need to further their work?

---

> > ### Comment · Reviewer_kE2w · 2024-11-25
> >
> > Thank you for your response. Most of my concerns have been resolved. Therefore, I have adjusted my score accordingly.

---

> > > ### Author Response · Authors · 2024-11-27
> > >
> > > Thank you! Based on the comments we've made the respective changes to the manuscript as well -- please let us know if you have any more feedback !

---

### Meta-Review · Area_Chair_jonN · 2024-12-21

**Metareview:**

The paper introduces a method using Sparse Autoencoders (SAEs) to achieve efficient, interpretable topic alignment in LLMs by scoring neurons and modifying layer outputs, reducing training and inference costs.

Strengths:
- Reviewers acknowledged that this paper introduces a new application of SAEs for achieving efficient and interpretable topic alignment in LLMs.
- Reviewers agree that this work could be a potential method for making LLMs more interpretable and adaptable to real-world tasks.

Weaknesses:
- Scalability issue: the paper does not apply the proposed method to more modern LLMs of larger sizes, thus making the reader question the generalizability of the method.
- The performance/result varies significantly based on the SAE configuration.
- Most importantly, one of the core aspects of this paper is LLM interpretation done by the proposed method, but the paper fails to quantitatively compare the proposed method with other existing LLM interpretation techniques.

**Additional Comments On Reviewer Discussion:**

During the rebuttal, the authors addressed some of the core concerns raised by the reviewers such as lack of baseline comparisons (the authors added baselines such as LoRA and prompts based approaches), scalability of the proposed method to larger models (the authors made theoretical justification). But the concern regarding the failure to quantitatively compare of interpretability remained.

---

### Decision · Program_Chairs · 2025-01-22

Reject